# Association between Visceral Adipose Tissue Metabolism and Cerebral Glucose Metabolism in Patients with Cognitive Impairment

**DOI:** 10.3390/ijms25137479

**Published:** 2024-07-08

**Authors:** Mi-Hee Yu, Ji Sun Lim, Hyon-Ah Yi, Kyoung Sook Won, Hae Won Kim

**Affiliations:** 1Department of Nuclear Medicine, Keimyung University Dongsan Hospital, Daegu 42601, Republic of Korea; yumihee555@gmail.com (M.-H.Y.); lzsunny@hanmail.net (J.S.L.); won@dsmc.or.kr (K.S.W.); 2Department of Neurology, Keimyung University Dongsan Hospital, Daegu 42601, Republic of Korea; geschwind@dsmc.or.kr

**Keywords:** visceral adipose tissue metabolism, cerebral glucose metabolism, cognitive impairment, Alzheimer’s disease

## Abstract

Visceral adipose tissue (VAT) dysfunction has been recently recognized as a potential contributor to the development of Alzheimer’s disease (AD). This study aimed to explore the relationship between VAT metabolism and cerebral glucose metabolism in patients with cognitive impairment. This cross-sectional prospective study included 54 patients who underwent ^18^F-fluorodeoxyglucose (^18^F-FDG) brain and torso positron emission tomography/computed tomography (PET/CT), and neuropsychological evaluations. VAT metabolism was measured by ^18^F-FDG torso PET/CT, and cerebral glucose metabolism was measured using ^18^F-FDG brain PET/CT. A voxel-based analysis revealed that the high-VAT-metabolism group exhibited a significantly lower cerebral glucose metabolism in AD-signature regions such as the parietal and temporal cortices. In the volume-of-interest analysis, multiple linear regression analyses with adjustment for age, sex, and white matter hyperintensity volume revealed that VAT metabolism was negatively associated with cerebral glucose metabolism in AD-signature regions. In addition, higher VAT metabolism was correlated with poorer outcomes on cognitive assessments, including the Korean Boston Naming Test, Rey Complex Figure Test immediate recall, and the Controlled Oral Word Association Test. In conclusion, our study revealed significant relationships among VAT metabolism, cerebral glucose metabolism, and cognitive function. This suggests that VAT dysfunction actively contributes to the neurodegenerative processes characteristic of AD, making VAT dysfunction targeting a novel AD therapy approach.

## 1. Introduction

Alzheimer’s disease (AD), the most prevalent neurodegenerative disease, is characterized by progressive cognitive decline and the accumulation of amyloid-β (Aβ) plaques and tau tangles in the brain [1]. Although the exact cause of AD has not been confirmed, its etiology is multifactorial and involves a complex interplay of genetic, environmental, and lifestyle factors [2,3]. Recent studies have highlighted the role of metabolic disorders as significant risk factors for AD, indicating a complex relationship between systemic metabolic dysfunction and neurodegenerative processes [4]. This underscores the critical need to explore metabolic pathways related to neurodegeneration, given the established connections between chronic conditions such as diabetes, cardiovascular diseases, and AD [5,6].

Visceral adipose tissue (VAT), traditionally considered an energy store, is now acknowledged for its active endocrine role, which affects systemic health. Recent studies suggest that VAT dysfunction significantly contributes to the development of AD by exacerbating systemic inflammation and metabolic dysregulation, both of which are closely linked to neurodegenerative diseases [7,8]. The secretion of pro-inflammatory cytokines and adipokines from dysfunctional VAT has been implicated in the promotion of neuroinflammation, which may drive the neuropathological processes characteristic of AD [9]. Additionally, recent investigations suggest a correlation between VAT dysfunction and an increased cerebral Aβ burden, suggesting that VAT dysfunction might directly influence the pathogenesis of AD by exacerbating cerebral amyloidogenesis and tau pathology [10,11]. These findings suggest that VAT dysfunction is associated not only with metabolic syndrome but also with AD [12,13]. Despite these connections, a definitive link between VAT dysfunction and neurodegenerative processes in AD is yet to be firmly established. Given that neurodegeneration is often associated with an increase in Aβ burden, a clear correlation between VAT dysfunction and neurodegeneration could provide compelling evidence that VAT dysfunction contributes to the development and progression of AD.

2-Deoxy-2[fluorine-18]-fluoroglucose (^18^F-FDG) positron emission tomography/computed tomography (PET/CT) has proven to be an invaluable tool for simultaneously assessing VAT metabolism, indicative of VAT function, and quantifying cerebral glucose metabolism, indicative of cerebral function [14,15]. ^18^F-FDG PET/CT has been shown to be useful in evaluating VAT function with various systemic conditions, including metabolic syndrome, cardiovascular disease, and cancer, highlighting the clinical utility of this imaging modality in assessing VAT dysfunction [16,17,18]. By leveraging ^18^F-FDG PET/CT, this study aimed to explore the relationship between VAT metabolism and cerebral glucose metabolism in patients with cognitive impairment, potentially confirming the role of VAT dysfunction in AD progression and enhancing our understanding of the underlying mechanisms driving this devastating disease.

## 2. Results

### 2.1. Population Characteristics

Among the 101 eligible patients who attended our memory clinic for cognitive function assessment, those with a Mini-Mental State Examination (MMSE) score below 10, those who withdrew consent, those diagnosed with cognitive impairment resulting from conditions distinct from AD, and those currently suspected of acute infection or inflammatory states were systematically excluded from the study (Figure 1). This study included a total of 54 subjects (age: 66.4 ± 8.4 years; female, 34 (63.0%)). Among the total cohort, 28 subjects (51.9%) exhibited pathologic changes related to Alzheimer’s. Clinically, 18 were diagnosed as cognitively unimpaired (CU), 14 as having mild cognitive impairment (MCI), and 22 as having dementia. Specifically, among the 18 CU patients, 5 (27.8%) had Alzheimer pathology, as did 7 (50.0%) of the 14 MCI patients and 14 (63.6%) of the 22 dementia patients. The results of the neuropsychological assessment in the total cohort, including cognitively unimpaired, mild cognitive impairment, and dementia subjects, are summarized in Appendix A. The subjects were categorized into two groups based on VAT metabolism, using a cutoff value of 0.71, which represented the mean maximum standardized uptake value (SUV_max_) in the total cohort: 27 in the low- and 27 in the high-VAT-metabolism group. The SUV_max_ of the VAT in the high-VAT-metabolism group was significantly higher than that in the low-VAT-metabolism group (0.8 ± 0.1 vs. 0.6 ± 0.1, *p* < 0.001). No significant differences were observed between the low- and high-VAT-metabolism groups in terms of sex; body mass index (BMI, kg/m^2^); years of education; prevalence of diabetes, hypertension, cardiovascular disease, and hyperlipidemia; or white matter hyperintensity (WMH) volume (Table 1). Cognitive function, evaluated using the Seoul Neuropsychological Screening Battery II (SNSB II), was significantly associated with age, education level, and WMH volume but not with sex, BMI, diabetes, hypertension, cardiovascular disease, or hyperlipidemia. All of the neuropsychological assessment scores except RCFT copy and COWAT were significantly associated with AD-signature region SUVR (Appendix A).

### 2.2. Association of VAT Metabolism with Cerebral Glucose Metabolism

Cerebral glucose metabolism was compared between the high- and low-VAT-metabolism groups based on voxel-wise analysis using statistical parametric mapping (SPM). This analysis revealed significantly decreased cerebral glucose metabolism in the high-VAT-metabolism group compared to the low-VAT-metabolism group across the bilateral parietal and temporal cortices (Figure 2). Table 2 shows the specific brain regions in which the high-VAT-metabolism group exhibited decreased cerebral glucose metabolism relative to the low-VAT-metabolism group.

In addition to voxel-wise analysis, the association between VAT metabolism and the AD-signature region SUVR, defined as cerebral glucose metabolism in regions affected by AD, was evaluated using volume of interest (VOI) analysis. In this VOI analysis, the AD signature SUVR was significantly lower in the high-VAT-metabolism group than in the low-VAT-metabolism group (0.96 ± 0.05 vs. 0.93 ± 0.05, *p* = 0.016) (Appendix A). Furthermore, Pearson’s correlation analyses showed that VAT SUV_max_ was negatively correlated with the AD-signature region SUVR (r = −0.340, *p* = 0.012). Multiple linear regression analysis, adjusted for age, sex, and WMH volume, revealed that VAT SUV_max_ was significantly associated with the AD-signature region SUVR (*β* = −0.252, *p* = 0.047). Detailed information regarding the association between VAT metabolism and the SUVR of the AD-signature region is presented in Appendix A.

### 2.3. Association between VAT Metabolism and Cognitive Function

The relationship between cognitive function and VAT SUV_max_ across the cohort was analyzed using both univariate and multivariate models (Table 3). In the univariable analysis, VAT SUV_max_ showed significant negative correlations with various cognitive assessments (Korean version of the Boston Naming Test (K-BNT) (r = −0.297, *p* = 0.034), Rey Complex Figure Test (RCFT) immediate recall (r = −0.302, *p* = 0.047), and Controlled Oral Word Association Test (COWAT) for supermarket items (r = −0.396, *p* = 0.010)), indicating an initial association between higher VAT metabolism and poorer cognitive performance. Subsequent multivariate analysis, which controlled for age, sex, BMI, education, and WMH volume, further substantiated these findings. Specifically, VAT SUV_max_ was significantly and negatively associated with the K-BNT (*β* = −0.271, *p* = 0.012), COWAT for supermarket items (*β* = −0.275, *p* = 0.025), phonemic fluency (*β* = −0.253, *p* = 0.024), Stroop Color Reading Test (*β* = −0.243, *p* = 0.034), and the Korean Mini-Mental State Examination (K-MMSE) score (*β* = −0.183, *p* = 0.041). These results highlight the significant negative correlation between VAT SUV_max_ and performance in specific cognitive domains, particularly naming ability, verbal fluency, and potentially visual memory and executive function, after adjusting for key demographic and clinical variables.

## 3. Discussion

Increased VAT metabolism, as measured using ^18^F-FDG PET/CT, has been demonstrated to indicate VAT dysfunction [16,19,20]. This study investigated the relationships between VAT metabolism, cerebral glucose metabolism, and cognitive function in individuals with cognitive impairment. Our findings indicate a significant negative correlation between VAT metabolism and cerebral glucose metabolism, particularly in regions known to be affected by AD. Additionally, even after controlling for demographic and clinical variables, higher VAT metabolism was correlated with poorer performance on cognitive tests, including naming ability, verbal fluency, and aspects of memory and executive function. These findings suggest that VAT dysfunction influences neurodegeneration in AD-associated brain regions, underscoring its potential role in the progression of AD.

VAT dysfunction, characterized by the abnormal release of pro-inflammatory cytokines and adipokines, plays a pivotal role in systemic inflammation and metabolic disturbances, both of which are intricately linked to AD development [21,22]. In this study, we measured VAT metabolism using ^18^F-FDG PET/CT as an indicator of VAT dysfunction. This noninvasive approach to assessing VAT metabolism using ^18^F-FDG PET/CT offers an innovative method for investigating the association between VAT dysfunction and neurodegenerative processes [23,24]. The relevance of this approach is supported by several studies that have highlighted the significant relationship between altered VAT metabolism and various health conditions, thus establishing ^18^F-FDG uptake in VAT as a reliable marker of VAT dysfunction [16,25]. For instance, a prospective study using ^18^F-FDG PET showed that ^18^F-FDG uptake in neck adipose tissue was a strong predictor of cardiovascular risk in a cohort of 173 individuals [17]. Similarly, another study using ^18^F-FDG PET/CT reported that ^18^F-FDG uptake in the VAT was positively correlated with high-sensitivity C-reactive protein (hsCRP) levels and the SUV_max_ of immune-related organs, indicating an association between VAT metabolism and systemic inflammation [16]. Furthermore, recent research has shown that ^18^F-FDG uptake in VAT is positively correlated with adiponectin levels and inversely correlated with insulin resistance, further substantiating the role of VAT metabolism as a proxy for assessing VAT dysfunction [15].

To substantiate the evidence that VAT dysfunction influences the development and progression of AD, our previous research established a significant correlation between increased VAT metabolism and elevated cerebral amyloid beta burden. These findings suggest that VAT dysfunction may contribute to its pathogenesis [10]. Furthermore, this study revealed a significant decrease in cerebral glucose metabolism in the high-VAT-metabolism group compared with the low-VAT-metabolism group, particularly across the parietal and temporal cortices. These regions are critical in AD, with impairments often indicating the onset and progression of the disease. The delineation of specific brain regions exhibiting decreased glucose metabolism in the high-VAT-metabolism group, as detailed in our results, not only supports the notion of a metabolic component of non-specific neurodegeneration but also indicates a possible direct or indirect influence of VAT dysfunction on AD pathology. This connection is substantiated by a broader research landscape. Clinical studies have demonstrated that increased VAT volume and associated metabolic dysfunction are linked with hallmark AD biomarkers, such as elevated leptin levels and decreased Aβ1-42 concentrations in the CSF [13,26,27]. Similarly, serum adiponectin levels have been shown to be higher in AD patients, potentially acting as a compensatory response to AD-related signaling deficits [13]. This suggests that higher adiponectin levels in AD may counteract the inflammatory and metabolic disturbances characteristic of the disease, potentially reflecting its neuroprotective properties [28]. Animal models have further elucidated this link, showing that high-fat-diet-induced VAT dysfunction can lead to increased levels of amyloid precursor protein (APP) in both adipose tissue and the brain, indicating a direct pathophysiological pathway that may contribute to AD development [29]. Recent clinical studies have investigated the correlation between VAT and AD [30]. A prospective study conducted on asymptomatic middle-aged adults with risk factors for AD revealed a significant association between VAT accumulation and the decline in brain Aβ burden [31]. Another recent study explored whether regional fat deposits, rather than central obesity, should be used to understand the mechanism underlying the association between adiposity and the brain and found that different regional fat deposits are likely associated with an increased risk of neurodegeneration and dementia, especially AD. These regional fat depots, including VAT, SAT, and hepatic fat, have been linked to cortical thinning, WMHs, reduced cerebral volumes, and smaller hippocampal volumes, the latter being one of the first regions affected by AD [32]. Collectively, these insights underscore VAT dysfunction’s critical role in cerebral metabolic and amyloidogenic processes central to AD, advocating systemic metabolic function as a key component of a holistic approach in understanding AD pathology and intervention.

Our study elucidates the critical link between increased VAT metabolism and cognitive decline, presenting a significant association with decreased performance in cognitive domains that are vulnerable to AD, such as naming ability, verbal fluency, and executive function. By revealing negative correlations between VAT metabolism and outcomes in cognitive tests, namely the K-BNT, COWAT for supermarket and phonemic categories, Stroop Color Reading Test, and K-MMSE, our research aligns with previous findings that metabolic dysfunctions, including those indicated by VAT metrics, are intricately linked to the pathogenesis of AD. These cognitive assessments are affected in AD, underscoring the relevance of our findings in identifying metabolic dysfunction as a contributing factor to neurodegenerative processes. Another study has revealed that visceral adipose NLRP3 in obesity impairs cognition via IL-1R1 in CX3CR1+ cells, leading to neuroinflammation and cognitive deficits. Activation of IL-1R1 in CX3CR1-expressing cells due to activation of the visceral adipose inflammasome results in deficits in hippocampus-dependent memory [33]. A recent study reported that the abnormal secretion of adipose factors, including inflammatory cytokines such as TNF-α, IL-6, and IL-1β, from VAT might penetrate the blood–brain barrier, leading to brain damage and cognitive decline [34,35]. Impaired vascular function from metabolic syndrome, often associated with VAT accumulation, may mediate the association between VAT and cognitive function [32]. This implies that VAT dysfunction could affect cognitive impairment not only in AD but also in cases of vascular dementia. Understanding the role of VAT in cognitive function could help to develop targeted interventions and personalized treatments for cognitive decline and dementia-related brain changes [32].

VAT metabolism increases can significantly impact the development and progression of Alzheimer’s disease (AD). Thus, controlling VAT metabolism could potentially mitigate the inflammatory and metabolic pathways that contribute to the progression of AD. Lifestyle interventions, such as diet and exercise, are foundational strategies. A Mediterranean diet, rich in anti-inflammatory foods like fruits, vegetables, and healthy fats, has been shown to reduce VAT and improve metabolic health [36]. Regular physical activity, particularly aerobic exercise, can decrease VAT mass and enhance insulin sensitivity, which may indirectly benefit cognitive function [37]. Pharmacological interventions also offer promising avenues. Metformin, a widely used antidiabetic drug, has been noted for its effects on reducing VAT and improving systemic inflammation and insulin sensitivity [38]. Thiazolidinediones, another class of antidiabetic medication, target peroxisome proliferator-activated receptor-gamma (PPAR-γ) and have been shown to reduce VAT and exert anti-inflammatory effects [39]. Additionally, GLP-1 receptor agonists, used in the management of diabetes, have demonstrated potential in reducing VAT and modulating inflammatory responses [40]. Furthermore, recent research highlights the role of the gut microbiome in regulating VAT metabolism and inflammation. Probiotics and prebiotics that modulate the gut microbiota composition could be another strategy to manage VAT and its systemic effects [41].

This study, while offering significant insights into the relationship between VAT dysfunction and neurodegeneration, has several limitations. The utilization of ^18^F-FDG uptake in VAT as a proxy for VAT function, despite the widespread acceptance of ^18^F-FDG PET/CT for assessing organ function, lacks direct histopathological validation linking ^18^F-FDG uptake to specific degrees of VAT dysfunction. Furthermore, the cross-sectional nature of this study precludes definitive causal inferences regarding the impact of VAT metabolism on cerebral glucose metabolism and AD progression. The modest sample size further constrains the extrapolation of our findings to broader populations. Despite these limitations, our study makes a pivotal contribution to the field by offering a novel perspective on the metabolic underpinnings of neurodegeneration. The application of ^18^F-FDG PET/CT to explore this link represents a methodological strength, providing a noninvasive means to assess metabolic activity and its association with cognitive decline. Future longitudinal studies, ideally with larger cohorts, are essential to corroborate our findings and deepen our understanding of the role of VAT in neurodegenerative diseases, reinforcing the potential for metabolic interventions in the prevention and management of cognitive impairment and AD.

## 4. Materials and Methods

### 4.1. Study Design and Participants

This prospective study included a sequential cohort of individuals who attended our memory clinic for cognitive function assessment between June 2015 and January 2017. Designed as a cross-sectional analysis, the inclusion criteria were as follows: (1) individuals of either sex, between the ages of 50 and 90 years; (2) those who underwent a comprehensive imaging assessment within four weeks of their clinic visit, including volumetric 3-Tesla brain magnetic resonance imaging (MRI), ^18^F-florbetaben (^18^F-FBB) brain positron emission tomography (PET), torso ^18^F-fluorodeoxyglucose (^18^F-FDG) PET, and brain FDG PET; (3) participants who completed a neuropsychological assessment using the SNSB II; and (4) subjects with thoroughly documented clinical data, including age, gender, body mass index (BMI), educational level, and medical history of diabetes, hypertension, hyperlipidemia, and cardiovascular disease. Exclusion criteria were set to exclude individuals with a Mini-Mental State Examination (MMSE) score < 10, those who withdrew consent, those diagnosed with cognitive impairments due to conditions other than AD (e.g., vascular dementia, hydrocephalus, and acute infection), or those suspected of having current acute infectious or inflammatory states, as evidenced by CT or PET scans. Additionally, individuals taking medications that could affect VAT metabolism, such as antidiabetic and anti-inflammatory drugs, within 24 h prior to PET scans were excluded.

All patients were classified according to syndromal cognitive staging combined with biomarkers based on the 2018 National Institute on Aging-Alzheimer’s Association (NIA-AA) Research Framework: cognitively unimpaired (CU), mild cognitive impairment (MCI), and dementia with or without Alzheimer pathology [42]. For the evaluation of pathologic changes related to Alzheimer’s, composite ^18^F-FBB standardized uptake value ratios (SUVR_FBB_) were calculated from ^18^F-FBB brain PET scans, as previously described [43]. Patients with a composite SUVR_FBB_ of ≥1.39, a cut-off value that reflects an abnormally high cerebral Aβ burden, were considered positive for amyloid-β (Aβ). Conversely, patients with a composite SUVR_FBB_ of <1.39 were considered negative for Aβ [43]. This study conformed to the ethical guidelines of the Declaration of Helsinki and was approved by the Institutional Review Board of Dongsan Hospital (Approval No. 2018-02-011). Written informed consent was obtained from all participants or their legal guardians.

### 4.2. ^18^F-FDG Torso PET/CT Imaging

Torso ^18^F-FDG PET/CT was performed using a state-of-the-art Biograph mCT-64 system (Siemens Healthcare, Knoxville, TN, USA). To minimize variations in glucose metabolism and any potential interference with ^18^F-FDG uptake, ^18^F-FDG PET/CT imaging was conducted with standardized fasting (participants fasted for at least six hours before the scan), consistent timing (scans were scheduled in the morning), a controlled environment (maintaining a calm and stress-free setting), restricted physical activity (vigorous physical activity was restricted for 24 h prior to the scan), and maintaining blood glucose levels below 150 mg/dL. Following a tailored intravenous injection of 4.0 MBq/kg of ^18^F-FDG, dedicated PET images were systematically acquired in a three-dimensional acquisition mode approximately 50–60 min post-injection, allowing for optimal tracer distribution and uptake analysis. To complement PET imaging and facilitate accurate attenuation correction, non-enhanced low-dose CT scans were performed concurrently. These CT scans utilized a spiral scanning mode configured at 120 kVp and 150 mAs, employing the True X enhancement algorithm to improve image quality without significantly increasing radiation exposure.

To determine the degree of VAT metabolism, ^18^F-FDG uptake in the VAT was measured using a dedicated PET workstation (AdvantageWorkstation 4.3) on torso ^18^F-FDG PET/CT images, as previously described [17]. VAT was defined as intra-abdominal adipose tissue identified on CT images using predefined Hounsfield units (ranging from 70 to 110 HUs). ^18^F-FDG uptake in the VAT was quantified by drawing a region of interest (ROI) around each VAT on a CT slice, which led to the consistent generation of the same ROIs on the transaxial PET images. ROIs were drawn on each slice of the three VAT areas in the right colic, left colic, and sigmoid mesenteries. The standardized uptake value (SUV) ratio was calculated as follows: SUV = tracer activity in the ROI (MBq/mL)/injected dose (MBq)/total body weight (g). SUV_max_ was defined as the highest SUV within the ROI, and VAT SUV_max_ was defined as the average SUV_max_ in the three VAT areas. The VAT metabolism status was divided by the mean value of VAT SUV_max_ measured in the total cohort in a previous study [10]: subjects with VAT SUV_max_ < 0.71 were classified as the low-VAT-metabolism group, while subjects with VAT SUV_max_ ≥ 0.71 were classified as the high-VAT-metabolism group.

### 4.3. ^18^F-FDG Brain PET/CT Imaging

Brain ^18^F-FDG PET/CT imaging followed a similar protocol to torso imaging and was conducted within 40 to 50 min post-injection on the same day, before the ^18^F-FDG torso PET/CT scan. A specialized light foam rubber holder was employed to ensure that the participant’s head remained stable throughout the scan, thereby minimizing movement artifacts. Similar to torso imaging, non-enhanced low-dose CT scans were performed for attenuation correction and accurate localization of the cerebral ^18^F-FDG uptake.

For imaging analysis, we used SPM12 (Well Trust Center for Neuroimaging, London, UK) and MATLAB (R2018a, The MathWorks Inc., Natick, MA, USA) to conduct a voxel-based examination of the MRI and PET scans. The co-registration of these images was achieved using a mutual information algorithm, followed by spatial normalization for alignment with the whole brain. Subsequently, the images were smoothed using an 8 mm isotropic Gaussian filter for analysis uniformity. We performed voxel-wise t-statistics to identify the regions showing significant decreases in ^18^F-FDG uptake, setting the threshold for significance at *p* = 0.005 (uncorrected) with a minimum cluster size of 100 voxels. Areas with significant changes were then mapped onto a 3D brain model or high-resolution MRI template provided by SPM12 for accurate anatomical identification, with coordinates converted from MNI to Talairach space for precise localization.

To quantify cerebral glucose metabolism, volume of interest (VOI) analysis was conducted using the PMOD software program (PMOD Technologies Ltd., Zurich, Switzerland) as previously described [44]. The VOIs were individually defined on ^18^F-FDG PET images in the central region: the lateral, medial, and orbital frontal cortices; the lateral temporal cortex; the lateral and medial parietal cortices; the lateral and medial occipital cortices; and the limbic lobe. Standardized ^18^F-FDG uptake values were obtained from the defined regional VOIs, and the regional standardized ^18^F-FDG uptake value ratio (SUVR) was calculated by dividing the standardized ^18^F-FDG uptake value for the individual target region by that for the cerebellum [45]. The AD-signature region SUVR is defined as the weighted mean of the bilateral temporal, parietal, and insular cortices and limbic lobes, which are the regions specifically affected by AD [46].

### 4.4. Neuropsychological Assessment

All participants underwent a condensed version of the Seoul Neuropsychological Screening Battery II (SNSB II), focusing on critical cognitive domains. The assessment included the Digit Span Test for memory and attention, the Korean-Boston Naming Test for language (K-BNT), the Rey–Osterrieth Complex Figure Test (RCFT) for visuospatial skills and memory, he Seoul Verbal Learning Test (SVLT) for verbal memory, the Controlled Oral Word Association Test (COWAT) for verbal fluency, and the Stroop Color and Word Test for attention and cognitive flexibility. This streamlined battery aimed to provide a comprehensive overview of cognitive function, aiding in the diagnosis and understanding of cognitive impairments associated with neurodegenerative conditions.

### 4.5. Statistical Analyses

Statistical analyses were performed using IBM SPSS Statistics version 28. Continuous variables such as demographics, BMI, cerebral glucose metabolism, and cognitive function test scores were analyzed using means ± standard deviations and compared between the low- and high-VAT-metabolism groups using two-sample *t*-tests. Categorical variables, including sex and medical history, were evaluated using the chi-squared test. The relationship between VAT metabolism (assessed using SUV_max_) and cerebral glucose metabolism was examined using voxel-based analysis with SPM for cerebral metabolism. Furthermore, the relationship between VAT metabolism and cerebral glucose metabolism, as evaluated by VOI analysis, and cognitive function test scores were assessed using Pearson’s correlation test. Variables showing significant correlations were further analyzed using regression models to assess the impact of VAT metabolism on cerebral glucose metabolism or cognitive function. Statistical significance was set at *p* < 0.05.

## 5. Conclusions

Our study revealed a significant relationship between increased VAT metabolism, decreased cerebral glucose metabolism, and impaired cognitive function. This supports the hypothesis that VAT dysfunction actively contributes to the neurodegenerative processes characteristic of AD, suggesting that targeting VAT dysfunction could offer a novel approach for AD therapy. Future research is crucial to deepen our understanding of VAT’s role in neurodegeneration and to explore the therapeutic potential of modulating VAT function in the management of AD.

## Figures and Tables

**Figure 1 ijms-25-07479-f001:**
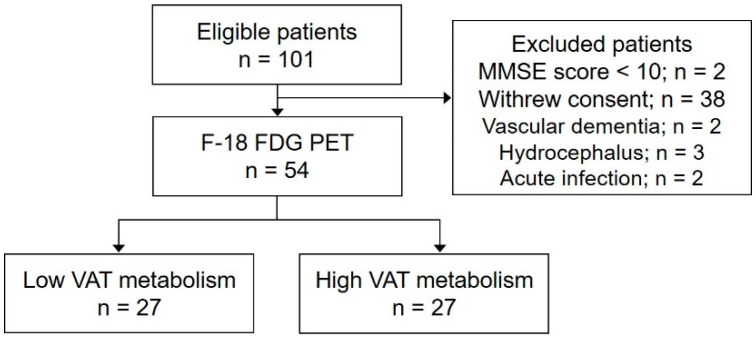
Flow diagram of the study population.

**Figure 2 ijms-25-07479-f002:**
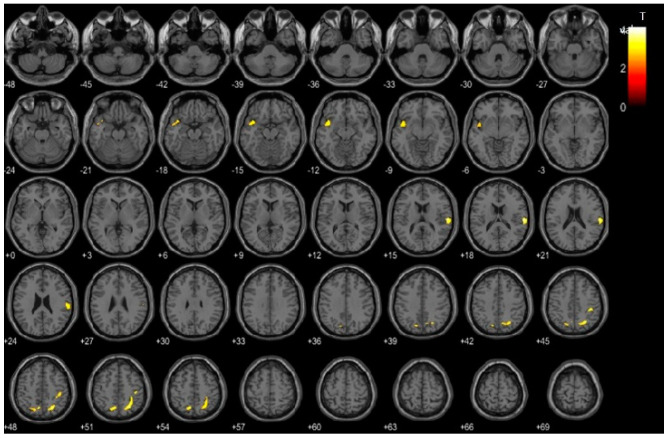
Voxel-based comparison of cerebral glucose metabolism between the high- and low-visceral adipose tissue (VAT)-metabolism groups. The statistical parameter mapping t-maps were su-perimposed on the volume-rendered magnetic resonance imaging (MRI) and T1-weighted template in the axial plane for the high-VAT-metabolism group < low-VAT-metabolism group (*p* < 0.005, uncorrected at voxel level, cluster size > 100 voxels).

**Table 1 ijms-25-07479-t001:** Comparison of clinical variables between low- and high-VAT-metabolism groups.

Characteristics	Total (n = 54)	Low-VAT-Metabolism Group (n = 27)	High-VAT-Metabolism Group (n = 27)	*p*
Age, years (SD)	66.4 (8.4)	65.4 (8.1)	67.5 (8.8)	0.181
Sex, female, n (%)	34 (63.0)	16 (59.3)	18 (66.7)	0.291
Body mass index (SD)	23.3 (3.4)	23.6 (2.8)	23 (4.0)	0.263
Education, years (SD)	11.5 (6.1)	11.4 (6.0)	11.5 (6.3)	0.474
Diabetes, n (%)	8 (14.8)	4 (14.8)	4 (14.8)	0.500
Hypertension, n (%)	16 (29.6)	10 (37.0)	6 (22.2)	0.121
Cardiovascular disease, n (%)	6 (11.1)	4 (14.8)	2 (7.4)	0.198
Hyperlipidemia, n (%)	9 (16.7)	5 (18.5)	4 (14.8)	0.361
WMH volume (SD)	3.4 (5.1)	2.4 (3.1)	4.4 (6.4)	0.083
VAT SUV_max_	0.7 (0.2)	0.6 (0.1)	0.8 (0.1)	<0.001

SD, standard deviation; WMH, white matter hyperintensity; VAT, visceral adipose tissue; SUV_max_, maximum standardized uptake value.

**Table 2 ijms-25-07479-t002:** Regions with significantly increased cerebral glucose metabolism in the high-VAT-metabolism group compared to the low-VAT-metabolism group in the SPM analysis (*p* < 0.005, uncorrected, k = 100).

Regions	Brodmann Area	Size	MINI Coordinates			T Value	*p*
			X	Y	Z		
Right parietal lobe, postcentral gyrus	BA40	228	60	−22	20	4.14	<0.001
Right parietal lobe, superior parietal lobule	BA7	393	24	−62	50	3.70	<0.001
Right parietal lobe, superior parietal lobule	BA7		30	−52	52	3.35	0.001
Right parietal lobe, inferior parietal lobule	BA40		38	−32	46	3.30	0.001
Left parietal lobe, superior parietal lobule	BA7	161	−10	−64	52	3.65	<0.001
Left parietal lobe, precuneus	BA7		−12	−74	38	3.01	0.002
Left parietal lobe, superior parietal lobule	BA7		−26	−66	50	2.77	0.004
Left temporal lobe, superior temporal gyrus	BA38	182	−42	6	−10	3.36	0.001
Left temporal lobe, middle temporal gyrus	BA21		−48	4	−18	2.96	0.002

VAT, visceral adipose tissue; BA, Brodmann’s area.

**Table 3 ijms-25-07479-t003:** Association between neuropsychological variables and VAT metabolism in the overall cohort.

Cognitive Domain	Neuropsychological Variables	Univariable Model		Multivariable Model		
		r	*p*	Adjusted R^2^	Standardized *β* *	*p*
Attention	Digit span forward	−0.140	0.313	0.430	−0.145	0.178
Working Memory	Digit span backward	−0.182	0.191	0.416	−0.180	0.090
Language Ability	K-BNT	−0.297	0.034	0.684	−0.271	0.012
Visuospatial Ability	RCFT copy	−0.098	0.513	0.498	−0.170	0.138
Memory	SVLT immediate recall	−0.122	0.383	0.570	−0.135	0.182
Memory	SVLT delayed recall	−0.195	0.161	0.457	−0.206	0.068
Memory	SVLT recognition	−0.212	0.131	0.323	−0.225	0.071
Memory	RCFT immediate recall	−0.302	0.047	0.287	−0.239	0.097
Memory	RCFT delayed recall	−0.234	0.127	0.397	−0.156	0.266
Verbal Fluency	COWAT animal	−0.137	0.331	0.442	−0.161	0.163
Verbal Fluency	COWAT supermarket	−0.396	0.010	0.488	−0.275	0.025
Verbal Fluency	COWAT phonemic	−0.300	0.060	0.574	−0.253	0.024
Executive Function	Stroop color reading	−0.273	0.077	0.502	−0.243	0.034
Global Cognition	K-MMSE score	−0.232	0.091	0.619	−0.183	0.041

K-BNT, Korean Boston Naming Test; RCFT, Rey Complex Figure Test; SVLT, Seoul Verbal Learning Test; COWAT, Controlled Oral Word Association Test; K-MMSE, Korean-Mini-Mental State Examination. * Values represent the standardized linear regression coefficients (β) of the correlation between the visceral adipose tissue’s maximum standardized uptake value (SUV_max_) and neuropsychological test scores, after adjusting for age, sex, BMI, education, and white matter hyperintensity volume.

## Data Availability

Data are contained within the article.

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
