# Peer review of "Association between Visceral Adipose Tissue Metabolism and Cerebral Glucose Metabolism in Patients with Cognitive Impairment"

_ijms, 2024, doi:10.3390/ijms25137479_

Round 1

Reviewer 1 Report

Comments and Suggestions for Authors

This is certainly an interesting study, but it requires more revision to be considered for publication:

The study undoubtedly gives us interesting information on the association between visceral adipose tissue metabolism, cerebral glucose metabolism, and cognitive functions. In this regard, the authors refer to cognitive decline in terms of Alzheimer's Disease. In this case, they must specify which criteria they used to diagnose "probable" Alzheimer's Disease (DSM-5 or what else?).

Secondly, the authors must include a summary table about the results of the different cognitive domains explored, concerning which cognitive tests.

Therefore, the authors must insert a further table summarizing the associations between visceral adipose tissue metabolism, cerebral glucose metabolism, and cognitive functions, with relative significance.

Finally, for a greater understanding of the text, the paragraph relating to materials and methods must be inserted after the introduction and before the results.

Author Response

We would like to express our gratitude for the valuable comments and suggestions provided by the reviewers. These inputs have significantly improved the quality and clarity of our manuscript.

All revised sections in the manuscript are highlighted in red text for easy identification.

Thank you for your thorough review and constructive feedback.

Reviewer Comment 1: The authors refer to cognitive decline in terms of Alzheimer's Disease. In this case, they must specify which criteria they used to diagnose "probable" Alzheimer's Disease (DSM-5 or what else?).

  • Author Response 1: As suggested by the reviewer, we classified all patients according to syndromal cognitive staging combined with biomarkers based on the 2018 National Institute on Aging-Alzheimer's Association (NIA-AA) Research Framework. Among the total cohort, 28 subjects (51.9%) exhibited Alzheimer pathologic changes. Clinically, 18 were diagnosed as cognitively unimpaired (CU), 14 with mild cognitive impairment (MCI), and 22 with dementia. Specifically, among the 18 CU patients, 5 (27.8%) had Alzheimer pathology, as did 7 (50.0%) of the 14 MCI patients, and 14 (63.6%) of the 22 dementia patients. (Added to page 2, line 94 and page 10, line 367, references 40-41)

Reviewer Comment 2: The authors must include a summary table about the results of the different cognitive domains explored, concerning which cognitive tests.

  • Author Response 2: We added the results of the neuropsychological assessment across the total cohort, including cognitively unimpaired, mild cognitive impairment, and dementia subjects, and summarized them in Supplementary Table 1.

Reviewer Comment 3: The authors must insert a further table summarizing the associations between visceral adipose tissue metabolism, cerebral glucose metabolism, and cognitive functions, with relative significance.

  • Author Response 3: We apologize for the oversight in not including the table summarizing the associations between visceral adipose tissue metabolism and cognitive functions. We have now added Table 3, which details these associations (Added to page 6, line 188). Additionally, we have included the results of the associations between cerebral glucose metabolism and cognitive functions in Supplementary Table S2.

Reviewer Comment 4: For a greater understanding of the text, the paragraph relating to materials and methods must be inserted after the introduction and before the results.

  • Author Response 4: We greatly appreciate the reviewer’s comment. However, upon consulting with the editor, we were advised that the manuscript sections should adhere to the order specified in the "Instructions for Authors": Introduction, Results, Discussion, Materials and Methods, and Conclusions.

Reviewer 2 Report

Comments and Suggestions for Authors

The authors have submitted a research article regarding an evaluation by PET-CT-based image analysis of a possible visceral adipose tissue (VAT) metabolism changes and the possible relationship between VAT dysfunction and cognitive function among patients with Alzheimer’s disease (AD). They examined the cerebral glucose metabolism as well as cognitive function levels. The authors demonstrated that a VAT metabolism assessed by using SUV(max) negatively corelated with cognitive function, illustrating a hypothesis suggesting that regulators of VAT metabolism might affect symptoms of AD. This issue is of interest, and impact of their results is strong. My overall concern with the article describing the current available data regarding beneficial availability of the evaluation of VAT metabolism in order to evaluate AD, offer something substantial that helps advance our understanding of advanced diagnosis and then effective medicinal management available in clinic.

To strengthen authors’ perspectives, the authors are strongly recommended to add a discussion in detail regarding how to control VAT metabolism to manage neurodegerative disorder such as AD. The discussion of what existing medications can be used to achieve such control will provide readers with some great insight.

In addition, in page 5, line 136, the authors mentioned Table 3, but I could not find it out within a set of submission.

Author Response

We would like to express our gratitude for the valuable comments and suggestions provided by the reviewers. These inputs have significantly improved the quality and clarity of our manuscript.

All revised sections in the manuscript are highlighted in red text for easy identification.

Thank you for your thorough review and constructive feedback.

Reviewer Comment 1: To strengthen authors’ perspectives, the authors are strongly recommended to add a discussion in detail regarding how to control VAT metabolism to manage neurodegenerative disorder such as AD. The discussion of what existing medications can be used to achieve such control will provide readers with some great insight.

  • Author Response 1: As suggested by the reviewer, we have incorporated a detailed discussion on controlling VAT metabolism to manage neurodegenerative disorders such as Alzheimer's disease (AD) into the discussion section of the manuscript. This includes an exploration of existing medications that can be used to achieve such control, providing readers with valuable insights. Specifically, we discuss lifestyle interventions like the Mediterranean diet and aerobic exercise, and pharmacological options such as metformin, thiazolidinediones, and GLP-1 receptor agonists, along with emerging therapies targeting adipokines and the gut microbiome. (Added to pages 8-9, line 301, references 34-39)

Reviewer Comment 2: In addition, in page 5, line 136, the authors mentioned Table 3, but I could not find it out within a set of submission.

  • Author Response 2: We apologize for the oversight in not including the table summarizing the associations between visceral adipose tissue metabolism and cognitive functions. We have now added Table 3, which details these associations. (Added to page 6, line 188)

Reviewer 3 Report

Comments and Suggestions for Authors

The manuscript presents interesting data confirming the association between the visceral glucose metabolism and Alzheimer’s disease. The authors, using 18F-fluorodeoxy-14 glucose (18F-FDG) brain and torso positron emission tomography/computed tomography (PET/CT) demonstrated lower brain glucose metabolism in the group of patients with high visceral glucose metabolism as compared with the group of visceral high glucose metabolism in Alzheimer’s disease patients. They found also significant negative correlations between visceral glucose metabolism and cognitive functions.

The methodology, employed also in previous studies of this team, is ingenious and appropriate. The study design is proper. The presentation and discussion of results is clear and logical. Inclusion and exclusion criteria of the patients are reported. The conclusions are scientifically sound and do go beyond the experimental data presented.

An obvious limitation of the study is the low number of patients examined (54), as admitted in the Discussion.

Remarks:

Was the recent or systematic uptake of medicines which could affect glucose metabolism considered as an exclusion factor?

Were the patients examined in the same part of the day or was it technically not possible?

Have the authors checked how stable are the glucose metabolism parameters assessed with 18F-glucose (how big are day-to-day and diurnal variations)?

Author Response

We would like to express our gratitude for the valuable comments and suggestions provided by the reviewers. These inputs have significantly improved the quality and clarity of our manuscript.

All revised sections in the manuscript are highlighted in red text for easy identification.

Thank you for your thorough review and constructive feedback.

Reviewer Comment 1: Was the recent or systematic uptake of medicines which could affect glucose metabolism considered as an exclusion factor?

  • Author Response 1: Yes, the recent or systematic uptake of medications that could affect glucose metabolism was considered an exclusion factor. For FDG PET/CT imaging, antidiabetic medications were prohibited within 24 hours before the scan. Additionally, individuals with current acute infectious or inflammatory states were excluded, ensuring no participants were taking anti-inflammatory drugs. Therefore, in our study, individuals taking medications that could affect VAT metabolism, such as antidiabetic and anti-inflammatory drugs, were excluded. This exclusion criterion has been added to the methods section for clarity. (Added to page 10, line 363)

Reviewer Comment 2: Were the patients examined in the same part of the day or was it technically not possible?

  • Author Response 2: Brain FDG PET/CT imaging was obtained 10 minutes before the FDG torso PET imaging on the same day. However, the neuropsychological assessment and brain MRI were conducted on different days due to the irregular and densely packed schedules of other patients at the hospital, making it impossible to align all assessments on the same date. We have added more detailed information about the timing of FDG PET/CT scans to the methods section for clarity. (Added to page 11, line 423)

Reviewer Comment 3: Have the authors checked how stable are the glucose metabolism parameters assessed with 18F-glucose (how big are day-to-day and diurnal variations)?

  • Author Response 3: We did not directly assess the stability of glucose metabolism parameters. However, most studies report that the most stable period for glucose metabolism in FDG PET/CT imaging typically occurs 50 to 60 minutes after FDG injection, allowing for optimal tracer distribution and uptake analysis. Despite this, glucose metabolism can exhibit day-to-day and diurnal variations influenced by circadian rhythms, recent food intake, physical activity, and stress levels. To minimize these variations and obtain consistent results, we conducted FDG PET/CT imaging with standardized fasting (participants fasted for at least six hours before the scan), consistent timing (scans were scheduled in the morning), a controlled environment (maintaining a calm and stress-free setting), restricted physical activity (vigorous physical activity was restricted for 24 hours prior to the scan), and maintaining blood glucose levels below 150 mg/dL. This information has been added to the methods section. (Added to page 10, line 385)

Round 2

Reviewer 1 Report

Comments and Suggestions for Authors

The authors have responded comprehensively to the comments made upon the first revision of the manuscript. 

Author Response

We would like to extend our sincere thanks and appreciation to the reviewer for your insightful and constructive comments and suggestions on our manuscript.

Reviewer 2 Report

Comments and Suggestions for Authors

The authors have done a good job responding to reviewer comments and concerns in their revision. I believe the manuscript is significantly improved as a result. Now I recommend that this revised version of the manuscript can be accepted for publication in the journal IJMS.

Author Response

We would like to extend our sincere thanks and appreciation to the reviewer for your insightful and constructive comments and suggestions on our manuscript